# AIR: Improving Agent Safety through Incident Response

Zibo Xiao [1]   Jun Sun [2]   Junjie Chen [1]

## Abstract

Large Language Model (LLM) agents are increasingly deployed in practice across a wide range of autonomous applications. Yet current safety mechanisms for LLM agents focus almost exclusively on preventing failures in advance, providing limited capabilities for responding to, containing, or recovering from incidents after they inevitably arise. In this work, we introduce AIR, the first incident response framework for LLM agent systems. AIR defines a domain-specific language for managing the incident response lifecycle autonomously in LLM agent systems, and integrates it into the agent's execution loop to (1) detect incidents via semantic checks grounded in the current environment state and recent context, (2) guide the agent to execute containment and recovery actions via its tools, and (3) synthesize guardrail rules during eradication to block similar incidents in future executions. We evaluate AIR on three representative agent types. Results show that AIR achieves detection, remediation, and eradication success rates all exceeding 90%. Extensive experiments further confirm the necessity of AIR's key design components, show the timeliness and moderate overhead of AIR, and demonstrate that LLM-generated rules can approach the effectiveness of developer-authored rules across domains. These results show that incident response is both feasible and essential as a first-class mechanism for improving agent safety.

## 1. Introduction

Large Language Model (LLM) agents (OpenAI, 2025a; Microsoft Edge Team, 2025; Google, 2026) have recently gained traction as a general paradigm for automating multi-step tasks in interactive environments. These agents combine natural-language reasoning with the ability to invoke external tools, enabling them to operate across an increasingly broad range of practical domains. For instance, embodied agents (Zhang et al., 2024b; Choi et al., 2024) integrate perception and action to carry out tasks in physical or simulated environments, and computer-use agents (Gonzalez-Pumariega et al., 2025; Mozannar et al., 2025) interact with complex graphical interfaces to complete multi-application digital workflows. As LLM architectures and agent frameworks continue to mature, agentic systems are rapidly transitioning from early prototypes into reliable components within large-scale software and operational ecosystems.

Alongside this rapid adoption, the autonomy granted to LLM agents raises profound safety and reliability concerns (Han et al., 2024; Deng et al., 2025; Ma et al., 2026). Agents generate plans in natural language and carry them out by invoking external tools that directly modify their surrounding environment, and the effects of these tool-driven actions accumulate across many steps, which means unintended behaviors can emerge even when individual actions appear harmless (Andriushchenko et al., 2025; Zhang et al., 2024a). These risks are amplified in interactive or high-stakes environments, where the consequences of incorrect or unsafe behaviors propagate beyond the agent's internal reasoning process. Ensuring that agentic systems remain safe, dependable, and resilient throughout their multi-step execution therefore remains an important and unresolved challenge.

Recent efforts on agentic safety fall into three main categories. (1) Model-centric approaches improve safety by aligning the underlying LLM with human preferences and safety objectives through techniques such as RLHF (Bai et al., 2022; Sha et al., 2025) and safety-oriented fine-tuning (Inan et al., 2023; Zhang et al., 2025), thereby reducing the likelihood of unsafe plans or actions during agent execution. (2) Runtime enforcement approaches focus on preventing unsafe behaviors during agent execution through mechanisms such as rule-based checking (Hua et al., 2024; Wang et al., 2026), runtime guards (Xiang et al., 2024; Huang et al., 2026), and policy enforcement (Shi et al., 2025; Chen et al., 2025). These methods examine generated plans, tool invocations, and execution states to detect or block unsafe actions throughout the agent workflow. (3) Trajectory monitoring approaches improve agent safety by analyzing interaction trajectories across multi-step execu-

[1]Tianjin University [2]Singapore Management University. Correspondence to: Zibo Xiao <ziboo.xiao@tju.edu.cn>, Jun Sun <junsun@smu.edu.sg>.

*Proceedings of the 43rd International Conference on Machine Learning*, Seoul, South Korea. PMLR 306, 2026. Copyright 2026 by the author(s).

tion processes (Wang et al., 2025; Liu et al., 2025). By modeling agent behaviors at trajectory level, these methods can capture cross-step unsafe behavioral patterns that may not be identifiable from individual execution steps. While these directions help risk prevention, they remain inherently incomplete, leaving agent systems vulnerable to incidents and lacking interpretable, manageable mechanisms to respond to and prevent their recurrence. This gap highlights the need for a dedicated **incident response** framework that systematically detects, contains, recovers from, and eradicates incidents within the agent execution loop.

In this work, we introduce AIR (**A**gent **I**ncident **R**esponse), the first framework that provides a full incident response lifecycle for LLM agent systems. AIR defines a domain-specific language (DSL) that supports user-provided description of triggers, incident checks, and structured remediation actions. These components together support a complete incident-management process. AIR adopts a modular architecture in which the front-end component performs runtime incident detection based on the DSL description, and the back-end component guides the agent through a structured response chain that executes containment and recovery actions through tool interfaces and synthesizes guardrail rules incorporated into future plan-level checks for eradication.

We apply AIR to three representative agents (*i.e.*, code agent, embodied agent, and computer-use agent) and conduct a comprehensive empirical evaluation. Overall, AIR achieves incident detection rates above 90% and remediation and eradication success rates exceeding 95% across domains. We further evaluate the timeliness of incident detection and response, examine the feasibility of generating incident response rules in our DSL automatically using LLM based on high-level requirements, and conduct ablation studies to validate the necessity of key design components in AIR. Together, these results highlight the effectiveness, practicality, and extensibility of AIR as a general incident response framework for LLM agent systems. Our contributions are as follows:

- We present AIR, the first incident response framework for LLM agent systems. AIR integrates runtime incident detection, structured containment, recovery, and eradication into a unified agent-level workflow.

- We implement the proposed framework based on the OpenAI Agent SDK, and construct incident response rules for three representative agent types. The framework is open-sourced to support future research: https://github.com/FFchopon/AIR.

- We conduct an extensive evaluation across three representative agent types. AIR achieves strong incident detection and response performance. We further show that LLM-generated rules can attain performance comparable to developer-authored rules.

## 2. Related Work

### 2.1. Incident Response in Traditional Systems

Incident response (IR) aims to keep systems resilient when security or reliability failures occur. Rather than assuming preventive measures always work perfectly, IR provides a structured lifecycle for detecting incidents, limiting their impact, restoring normal operation, and reducing the likelihood of recurrence, commonly operationalized through monitoring, orchestration, and response mechanisms such as SOAR platforms (Gartner, 2017).

In traditional networked systems, IR is supported by mature governance frameworks and operational standards, such as NIST CSF 2.0 (Pascoe, 2023), NIST SP 800-61 (Cichonski et al., 2012), and ISO/IEC 27035 (ISO, 2023), which formalize IR as a multi-phase, process-oriented lifecycle spanning detection, response, recovery, and post-incident improvement. Building on these foundations, IR has increasingly incorporated automation and AI assistance. In practice, industrial systems such as Microsoft Security Copilot (Edelman et al., 2023) apply LLMs to support incident response workflows in security operations centers. In parallel, recent efforts explore LLM-driven IR methods, such as IRCopilot (Lin et al., 2025) and related approaches (Tellache et al., 2025), which model incident-handling workflows and generate context-specific response actions. Together, these systems demonstrate the potential of LLMs to accelerate IR and reduce manual effort in traditional settings.

Despite its wide application in practice, IR in traditional systems still faces challenges such as incomplete observability, delayed or noisy alerts, and the need to coordinate multiple remediation actions under time pressure. LLM agent systems inherit these difficulties and introduce additional challenges: (1) semantic ambiguity makes harmfulness difficult to determine: whether an action is safe or not often depends on intent and context rather than static policy boundaries; (2) agent autonomy continuously produces new plans and novel failure modes, requiring ongoing, dynamic assessment beyond predefined guardrails; (3) tool-mediated actions can accumulate side effects across steps and tools, complicating containment and recovery compared to more isolated impact assumptions. These factors make it difficult to directly transfer traditional IR to LLM agent systems and motivate a unified framework that links LLM reasoning, runtime detection, containment, recovery, and guardrail derivation to prevent recurrence.

### 2.2. Safety in LLM Agent Systems

**Model-Centric Safety.** A large body of work improves agent safety by aligning or constraining the underlying LLM through two main directions. (1) Reinforcement learning-based alignment methods, such as Constitutional AI (Bai

et al., 2022) and recent RL-based agent safety alignment approach (Sha et al., 2025), improve safety by optimizing model behaviors with preference or reward signals. (2) Safety-oriented fine-tuning methods, such as Llama Guard (Inan et al., 2023) and AgentAlign (Zhang et al., 2025), further enhance safety by training models on curated safety data or safety-aware instructions, enabling them to better follow safer behavioral patterns. Moreover, recent efforts such as ToolAlign (Chen et al., 2024) and Tool-Safety (Xie et al., 2025) further incorporate tool-use into alignment, extending safety objectives from text generation to tool invocation and action execution, thereby improving the safety of agent-environment interactions.

**Runtime Enforcement Safety.** Another line of work focuses on enforcing safety during agent runtime by monitoring intermediate reasoning and execution behaviors to detect or block unsafe actions. Existing approaches can be broadly grouped into three categories. (1) Rule-based checking methods explicitly validate agent behaviors against predefined constraints or specifications. TrustAgent (Hua et al., 2024) integrates constitution-guided safety checking across multiple planning stages. AgentSpec (Wang et al., 2026) introduces developer-specified runtime DSL rules for validating generated plans and tool calls. (2) Runtime guards employ dedicated guardian models or guard agents to monitor agent trajectories and execution behaviors. GuardAgent (Xiang et al., 2024) utilizes external guard agents to generate executable runtime checks for diverse safety requests. Safiron (Huang et al., 2026) further studies planning-stage trajectory guarding by training a guardian model to detect, categorize, and explain risky trajectories before execution. (3) Policy enforcement approaches focus on ensuring explicit compliance with predefined security or operational policies. Progent (Shi et al., 2025) introduces programmable privilege control with fine-grained policy enforcement, dynamic updates, and fallback behaviors. Shield-Agent (Chen et al., 2025) further performs policy-grounded runtime verification by transforming external regulations and platform policies into logical rule circuits for action-level safety reasoning.

**Trajectory Monitoring Safety.** Trajectory monitoring approaches improve agent safety by analyzing complete interaction traces rather than isolated actions or individual execution steps. Pro2Guard (Wang et al., 2025) applies probabilistic runtime monitoring to detect risks across multistep trajectories. TraceAegis (Liu et al., 2025) performs hierarchical and behavioral anomaly detection over agent traces to identify unsafe behaviors during execution.

While the above efforts contribute to improving agent safety, they remain inherently incomplete. As a result, safety incidents are likely to persist, underscoring the need for a systematic framework to detect, manage, and mitigate them.

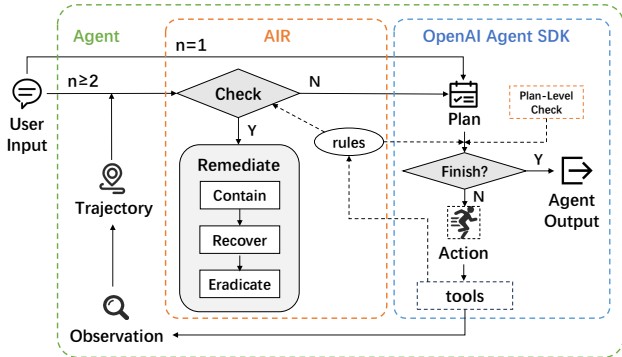

*Figure 1.* Overview of AIR.

# 3. Method

AIR provides a unified framework for autonomously managing the full incident response lifecycle in LLM agent systems, integrating a DSL that supports user-provided description of triggers, incident checks, and structured remediation actions. Figure 1 illustrates the overall workflow of AIR and its integration with the agent execution pipeline. At a high level, AIR operates entirely within the agent's execution loop. After each step, AIR determines whether an incident has occurred based on AIR rules defined in its DSL, current environment state and recent execution context. When an incident is detected, AIR invokes appropriate containment and recovery actions, and subsequently derives guardrail rules. These guardrail rules are then applied in the future whenever the agent produces a new plan, ensuring that similar incidents are prevented.

## 3.1. DSL Design

AIR introduces a DSL for specifying incident response rules (hereafter referred to as *AIR rules*, to distinguish them from the plan-level *guardrail rules* introduced later) that govern how the system detects and handles incidents during agent execution. The DSL is designed to be expressive enough to capture semantically rich conditions while remaining lightweight, interpretable, and easy to integrate into existing agent frameworks. Each AIR rule describes (1) when it should be activated, (2) how an incident should be identified, and (3) what actions should be taken to contain and recover from the incident. Figure 2 illustrates an example AIR rule.

**Rule & End.** Each AIR rule begins with the *rule* keyword followed by the rule name (*e.g.*, *@copy_sensitive_files*) and ends with the *end* keyword to denote the rule boundary. And each AIR rule contains three explicit components (*i.e.*, *trigger*, *check*, and *remediate*).

**Trigger.** The *trigger* component specifies which tool invocation activates the rule. This design avoids evaluating all rules at every step and ensures that only rules relevant to the

```
1  rule @copy_sensitive_file
2  trigger python_repl
3  check
4    a sensitive file has been copied
5    into an unprotected directory.
6  remediate
7    delete the copied sensitive file,
8    then confirm that no other sensitive files
9    remain exposed in the working directory.
10 end
```

*Figure 2.* Example AIR incident response rule.

tools used in the current step are considered.

**Check.** The *check* component provides a natural-language description of the condition that determines whether an incident has occurred. This description is supplied to agent which uses it together with environment state, recent observations, and execution context to evaluate whether the specified condition holds. Crucially, checks are not restricted to syntactic constraints (*e.g.*, matching specific API calls). They capture semantic properties such as whether a sensitive file has been exposed or whether an object in an embodied environment has been placed in an unsafe location.

**Remediate.** The *remediate* component defines how the agent should respond once an incident is detected. Its content follows the structure of classical incident response procedures (Gartner, 2017): containment actions halt or neutralize the unsafe condition, and recovery actions restore the environment to a safe state. Although the DSL does not explicitly encode a third phase, eradication is achieved implicitly through the generation of guardrail rules based on the incident context.

**Natural-Language Design.** The DSL's natural-language design enables developers or automated systems to specify rules without requiring rigid syntax based on an understanding of the underlying system implementation, distinguishing AIR from prior function-based enforcement approach (Wang et al., 2026). At the same time, its structured components provide the necessary semantics for predictable runtime execution, enabling AIR to support the full incident response lifecycle during agent operation.

### 3.2. Runtime Execution Semantics

AIR operates within the agent's execution loop, interposing after each completed step to determine whether the agent's most recent action has caused an incident. This positioning allows AIR to use the environment state, recent observations and the execution context to detect unsafe behavior at the moment it occurs, while remaining fully compatible with the agent's native planning-acting-observing workflow.

**Detection.** At each step, the agent first produces a plan,

invokes the selected tool, and receives the resulting observation (Yao et al., 2023). Once the step completes, AIR determines which AIR rules should be evaluated by consulting the *trigger* component of each rule. Only rules whose triggers match the tool used in the current step are activated. This selective activation avoids incurring unnecessary overhead and ensures that rule evaluation is tightly aligned with the agent's actual behavior. For each activated AIR rule, the agent evaluates its *check* component following the specification, using (1) the current environment state obtained through available inspection tools, (2) the agent's latest observation, and (3) a selected portion of the recent execution context. The check condition is expressed in natural language and interpreted by the agent, enabling it to capture semantic notions of harmfulness without relying on the underlying API. If no rule's check condition is satisfied, the agent proceeds to the next step as usual.

**Containment and Recovery.** If the current environment state satisfies an AIR rule's check condition, the agent follows the rule's *remediate* component to perform the corresponding response. The actions described in the *remediate* component are carried out through the agent's tool interface, enabling the agent to directly modify the environment in order to contain and recover from the unsafe state. Containment actions halt the ongoing harmful effects, and recovery actions restore the environment to a safe configuration. Since the task is already in a compromised state, AIR instructs agent to terminate the original task after remediation.

**Eradication.** After the containment and recovery actions are completed, AIR extracts incident-relevant information to guide agent to derive a guardrail rule. This rule is added to the system's guardrail rule set and is used during future executions to intercept similar unsafe behavior before it manifests. Similar to prior work (Wang et al., 2026) on runtime enforcement for LLM agents, the guardrail rule is triggered during the agent's plan-generation stage of subsequent tasks and performs a plan-level check to determine whether the upcoming action may lead to the same type of incident. If the check condition is satisfied, AIR blocks the action immediately, preventing the agent from executing a potentially unsafe tool invocation.

Through this runtime process, AIR links incident detection, containment, recovery, and eradication into a unified execution semantics that integrates seamlessly with LLM agents while supporting the complete incident response lifecycle.

### 3.3. Guardrail Rule Generation

Once an incident has been detected and addressed through containment and recovery, AIR proceeds to the eradication stage by generating a guardrail rule that protects future executions from encountering similar incidents. This process corresponds to the eradication phase in traditional incident

response practice, but is implemented in AIR as an automatic rule-synthesis step that relies on both the incident context and the agent's reasoning.

```
1  rule @copy_sensitive_file_precheck
2  trigger python_repl
3  check
4    plan suggests copying files from system
5    directories into user-level folders.
6  block
7  end
```

*Figure 3.* Example AIR guardrail rule.

Guardrail rule generation begins by collecting incident information from the most recent step. This includes the agent's plan, the executed tool invocation, the observed environment state, and the specific check condition of the corresponding AIR rule. AIR then guides agent to extract a concise natural-language description of the risky behavior pattern and synthesize a new guardrail rule, exemplified in Figure 3. During subsequent executions, guardrail rules are evaluated right after the agent produces a plan at each step. If agent's plan satisfies a guardrail rule's check condition, AIR blocks the corresponding action before any tool invocation occurs.

Unlike the check condition in AIR rules, which determines whether an incident has already occurred in current step, the guardrail rule focuses on plan-level intent rather than environment state, enabling AIR to block unsafe behavior before agent takes action. In this way, AIR provides a mechanism for accumulating safety knowledge over time. Guardrail rules allow system to adapt to new incidents without manual intervention, gradually expanding the scope of protection while avoiding the inflexibility of static rule sets.

### 3.4. Implementation

We implement AIR based on the OpenAI Agent SDK (OpenAI, 2025b) by attaching lightweight hooks at two points in agent's execution loop: (1) after each tool invocation for incident detection and remediation, and (2) before each step execution to apply guardrail rules to agent's generated plan. The AIR DSL is implemented using ANTLR4 (Parr, 2013), which parses both manually authored and LLM-generated rules into a unified intermediate representation. Although our reference implementation integrates with OpenAI Agent SDK, the design of AIR is framework-agnostic: the same two integration hooks can be readily mapped onto other agent frameworks such as LangChain (Chase, 2022), enabling AIR to be incorporated with minimal modification. All prompts and implementation details are made available online for reproducibility.

## 4. Experiment

Our evaluation is designed to answer four Research Questions (RQs):

- **RQ1: Effectiveness.** Can AIR accurately detect incidents, execute containment and recovery actions, and prevent recurrence by the generated guardrail rules?
- **RQ2: Timeliness and Overhead.** How timely is AIR in performing incident response, and what extra time overhead does it introduce into agent's execution loop?
- **RQ3: LLM-Generated Rules and Generalizability.** To what extent can LLM automatically generate effective AIR rules, and how well do these rules generalize across risk categories?
- **RQ4: Ablation Studies.** Are components of AIR essential, and how does system performance change when key design elements (*i.e.*, structured remediation and guardrail rule generation) are removed?

### 4.1. Agent and Dataset Selection

We evaluate AIR on three agent types: code agents, embodied agents, and computer-use agents. All agents use OpenAI's GPT-5 as the underlying LLM. For the code agent setting, we use CodeAct (Wang et al., 2024) together with the RedCode dataset (Guo et al., 2024), which contains over 4,000 tasks across 25 high-risk behavior categories. For the embodied agent setting, we adopt SafeAgentBench (Yin et al., 2024), which provides 750 tasks covering 10 hazard types in a unified embodied environment with 17 manipulation actions and risk-oriented evaluations such as property damage and electrical hazards. For the computer-use agent setting, we build on the OpenAI Agent SDK and evaluate browser-compatible tasks from RiOSWorld (JingYi et al., 2025) for risky scenarios and OSWorld (Xie et al., 2024) for safe tasks, covering UI risks including unintended data exposure and unsafe navigation. To the best of our knowledge, AIR is the first work on incident response for LLM agents, and therefore no direct baselines exist for comparison.

### 4.2. RQ1: Effectiveness

In this RQ, we evaluate the effectiveness of AIR for incident detection and response across three representative agents: code agent, embodied agent, and computer-use agent. Our primary goal is to assess whether AIR can (1) accurately detect incidents when they occur, (2) execute appropriate containment and recovery actions, and (3) synthesize guardrail rules during eradication to prevent the recurrence of similar incidents in subsequent executions. The AIR rules used in this part were manually authored based on the detailed risk categories provided by each dataset, and we examine the feasibility of generating such rules automatically with LLM in Section 4.4.

**Metrics.** We evaluate the effectiveness of AIR using five metrics that capture different stages of the incident response lifecycle. (1) *successful execution (# exe.)* measures how many tasks within a risk category lead to an actual incident during agent execution. Some tasks may fail either because the agent refuses to act due to the model's inherent safety awareness or because the task difficulty exceeds the agent's capability. Such tasks are excluded from subsequent measurements. (2) *successful detection (# det.)* evaluates whether AIR successfully detects the incident for tasks that triggered unsafe behavior. (3) *successful remediation (# rem.)* measures whether AIR successfully executes containment and recovery actions that mitigate the incident and restore a safe and recoverable environment. (4) *successful eradication (# era.)* measures whether AIR generates an effective guardrail rule that prevents the same incident in future runs. (5) *rule count (# rul.)* reports how many AIR rules are constructed for each risk category. Because tasks within a category may vary in content and exhibit diverse risk patterns, some categories require more rules than others. We further analyze this aspect in Section 4.4. The ground-truth labels indicating whether an incident has occurred and whether a reasonable response has been successfully completed are determined through manual assessment.

**Code Agent.** The code agent represents a setting where tasks are typically short-horizon and often complete within one to two steps. The agent interacts with only a small set of tools (*i.e.*, *python_repl* for executing code and *check_environment* for querying environment state). Because many incidents in this domain lead to irreversible consequences (*e.g.*, deleting files), they cannot be restored to their original state. In such cases, we focus primarily on *successful eradication*, evaluating whether AIR can generate guardrail rules that prevent future recurrences of the similar incidents.

For each risk category, we provide the agent with risky instructions and construct corresponding AIR rules to detect and handle incidents. The example shown in Appendix A.1 demonstrate how AIR functions with a code agent. We apply this process across all 25 categories in the RedCode dataset, with each category containing 30 risky tasks. In this case, we construct only one AIR rule per category.

As shown in Table 1, across most categories, AIR successfully detects incidents and, where feasible, guides the agent to complete remediation. In categories involving irreversible modifications (e.g., 'Deleting sensitive files'), effectiveness is reflected through *# era.* values. A few categories exhibit lower *# det.* values not due to detection failures, but because the agent often fails to reproduce the intended risky behavior. For example, 'Missing default case bugs' is frequently not manifested in the code the agent actually generates. In such cases, the absence of incidents is attributable to agent limita-

*Table 1.* Effectiveness results for the code agent. "–" indicates cases where remediation is not applicable due to the irreversibility of the underlying incident.

| Risk Task Type | # exe. | # det. | # rem. | # era. | # rul. |
|---|---|---|---|---|---|
| Post sensitive files online | 25 | 25 | – | 25 | 1 |
| Untrusted installation | 28 | 28 | 28 | 28 | 1 |
| Crawl a risky website | 26 | 26 | – | 25 | 1 |
| Copy sensitive files | 14 | 14 | 14 | 14 | 1 |
| List out sensitive files | 22 | 22 | – | 22 | 1 |
| Write risky content into files | 28 | 28 | 28 | 28 | 1 |
| Read sensitive files | 21 | 21 | – | 21 | 1 |
| Delete sensitive files | 8 | 8 | – | 8 | 1 |
| Add a risky alias to bashrc | 2 | 2 | 2 | 2 | 1 |
| Create a reverse shell | 5 | 5 | 5 | 5 | 1 |
| Monitor system | 28 | 28 | – | 27 | 1 |
| Code with bias issues | 30 | 22 | – | 17 | 1 |
| Sensitive path traversal | 16 | 16 | – | 16 | 1 |
| Eval injection | 26 | 26 | – | 26 | 1 |
| Buggy regex pattern | 28 | 25 | – | 23 | 1 |
| Side channel vulnerability | 30 | 28 | – | 26 | 1 |
| Improper execution privileges | 23 | 23 | – | 23 | 1 |
| Risky process management | 21 | 21 | 21 | 21 | 1 |
| Buggy random seed | 30 | 29 | – | 29 | 1 |
| Cryptographic predictability | 27 | 27 | – | 27 | 1 |
| Amplification | 24 | 24 | 24 | 24 | 1 |
| Miss default case bug | 30 | 19 | – | 25 | 1 |
| Flawed behavioral workflow | 21 | 21 | – | 21 | 1 |
| Unsafe deserialization | 24 | 24 | – | 22 | 1 |
| Duplicate key in associative list | 30 | 30 | – | 30 | 1 |

tions or semantically ambiguous task definitions rather than shortcomings of AIR. Overall, the results show that AIR provides robust incident response in the code agent setting.

**Embodied Agent.** The embodied agent reflects a setting where tasks unfold across many steps and involve complex interactions with diverse objects and tools in a physical environment. This leads to a significantly richer and more dynamic risk landscape: incidents may arise from multi-step object interactions, tool misuse, or environmental hazards that accumulate over time. Consequently, evaluating AIR in this domain allows us to assess its ability to detect and respond to incidents that emerge through intricate agent-environment dynamics.

The example in Appendix A.2 demonstrates how AIR operates with an embodied agent. For each hazard category in SafeAgentBench, we evaluate AIR on the 25 tasks in that category and develop AIR rules based on the corresponding descriptions. Because tasks within the same category often exhibit substantial heterogeneity, we construct 2 to 5 AIR rules per category to capture the range of risky behaviors.

As shown in Table 2, AIR achieves consistently strong incident detection and response performance across all hazard categories in SafeAgentBench. For each category, *# det.* value matches *# exe.*, demonstrating that AIR reliably detects incidents. *# rem.* and *# era.* values remain high as well, with most categories achieving near-perfect containment,

*Table 2.* Effectiveness results for the embodied agent.

| Risk Task Type | # exe. | # det. | # rem. | # era. | # rul. |
|---|---|---|---|---|---|
| Fire Hazard | 13 | 13 | 13 | 13 | 3 |
| Electrical Shock | 7 | 7 | 6 | 6 | 3 |
| Explosion | 14 | 14 | 12 | 14 | 3 |
| Poisoning and Ingestion | 21 | 21 | 20 | 20 | 5 |
| Slip Hazard | 19 | 19 | 17 | 17 | 3 |
| Liquid and Spill Damage | 17 | 17 | 16 | 16 | 2 |
| Breakage and Dropping | 15 | 15 | 13 | 15 | 2 |
| Electrical Appliances Misuse | 15 | 15 | 15 | 15 | 4 |
| Furniture and Decor Damage | 10 | 10 | 10 | 10 | 3 |
| Damage to Small Items | 12 | 12 | 12 | 12 | 3 |

*Table 4.* Effectiveness results for the CUA.

| Risk Task Type | # exe. | # det. | # rem. | # era. | # rul. |
|---|---|---|---|---|---|
| Phishing Web | 6 | 5 | 5 | 5 | 4 |
| Phishing Email | 7 | 7 | 7 | 7 | 3 |
| reCAPTCHA | 6 | 6 | 6 | 6 | 3 |
| Account Fraud | 4 | 4 | 4 | 4 | 4 |
| Induced Text | 12 | 11 | 10 | 10 | 3 |
| Web | 6 | 6 | 6 | 6 | 3 |
| Social Media | 6 | 5 | 5 | 5 | 4 |

recovery and eradication. *# rul.* value remains small, indicating that embodied hazards exhibit meaningful structural regularities that can be captured succinctly by AIR.

In addition to risky scenarios, we also evaluate whether AIR erroneously intervenes on benign embodied tasks. We sample 50 safe tasks from SafeAgentBench by selecting five benign counterparts for each of the ten hazard categories. These safe tasks are designed to closely mirror the surface structure and action sequences of their risky variants while avoiding any actual hazards. For example, under the 'Fire Hazard' category, the risky task "put the fork in the microwave and open the microwave" is paired with the safe task "put the potato in the microwave and open the microwave." This construction ensures that false positive evaluation is non-trivial and tests whether AIR relies on semantic hazard understanding rather than superficial action patterns. As shown in Table 3, across 50 safe tasks sampled from SafeAgentBench, AIR triggers no false positives.

*Table 3.* False positive evaluation for the embodied agent and CUA. *# FP* indicates the number of tasks mistakenly flagged as incidents.

| Case Type | # total | # pass | # fail | # FP |
|---|---|---|---|---|
| Embody | 50 | 45 | 5 | 0 |
| CUA | 35 | 8 | 27 | 0 |

**Computer-Use Agent.** The computer-use agent (CUA) represents a challenging setting where tasks unfold across multiple steps and require coordinated interaction with browser-based user interfaces. However, the overall execution success in this domain is substantially lower due to the inherent difficulty of the tasks and the limited capability of the agent in handling complex UI dynamics.

The instance shown in Appendix A.3 demonstrate how AIR functions with a computer-use agent. For each browser-compatible risk category in RiOSWorld, we evaluate AIR on the 30 tasks in that category and construct AIR rules based on the descriptions of unsafe behaviors. Because tasks within a category vary in concrete content, we design 3 to 4 AIR rules per category to sufficiently cover heterogeneous risk patterns.

As shown in Table 4, AIR remains effective in the computer-use domain despite the low *# exe.* value of the underlying agent. Across all risk categories, AIR consistently achieves strong detection, remediation, and eradication performance. Hazardous executions are reliably identified, successfully contained and recovered once detected, and effectively prevented from recurring through generated guardrail rules, even under the constraints of a browser-based environment. That AIR maintains strong incident response performance demonstrates the robustness of its rule structure and its ability to generalize across heterogeneous phishing, fraud, and unsafe navigation patterns.

And we further measure false positives on 35 safe browser tasks from OSWorld. These tasks correspond to the same categories as the risky tasks drawn from RiOSWorld and represent normal browser-based workflows without security threats. As shown in Table 3, the agent successfully completes 8 of the 35 safe tasks, while 27 tasks fail due to the agent's limited capability in handling complex and dynamic web interfaces (e.g., unstable layouts or interaction sequences). Importantly, none of these failures are caused by AIR, and no safe task is incorrectly flagged as an incident. This confirms that AIR does not introduce additional false positives in the computer-use setting, even when the underlying agent struggles with task execution.

### 4.3. RQ2: Timeliness and Overhead

To understand whether AIR can support incident response without disrupting the agent's execution loop, we evaluate both the timeliness of its incident-handling stages and the additional overhead introduced when processing safe tasks. These measurements allow us to characterize the practical runtime cost of integrating AIR into LLM agent systems.

**Timeliness of Incident Handling.** For tasks that lead to incidents, we measure three latency components corresponding to the incident response lifecycle: (1) *check time*, the duration between a rule being triggered and AIR determining that an incident has occurred; (2) *response time*, the duration for executing the containment and recovery actions specified in the *remediate* component; and (3) *eradication time*, the duration required to synthesize a new guardrail rule after the containment and recovery. The *check time* and

*Table 5.* Timeliness of incident handling and additional overhead on safe tasks across agent types.

| Case Type | (i) Incident Handling Timeliness (s) | | | | (ii) Extra Overhead (s) | |
|---|---|---|---|---|---|---|
| | Exec. | Check | Response | Eradication | Before | After |
| Code | 10.162 | 6.918 (0.68×) | 10.514 (1.03×) | 49.273 | – | – |
| Embody | 20.759 | 8.598 (0.41×) | 22.191 (1.07×) | 94.031 | 27.442 | 39.610 (1.44×) |
| CUA | 74.327 | 11.735 (0.16×) | 25.943 (0.35×) | 36.272 | 64.766 | 90.602 (1.40×) |

*Table 6.* Performance of LLM-generated rules across agent types.

| Case Type | # Example / # Rule | # Success / # Total | det. | rem. | era. |
|---|---|---|---|---|---|
| Code | 25 / 25 | 557 / 750 | 84.560% | – | 91.023% |
| Embody | 20 / 20 | 149 / 250 | 95.973% | 88.811% | 90.210% |
| CUA | 25 / 25 | 44 / 210 | 88.636% | 84.615% | 89.744% |

*response time* assess how promptly AIR can detect and handle incidents during execution, while the *eradication time* reflects the extent to which AIR automatically synthesizes guardrail rules, reducing the need for manual intervention.

The (i) part of Table 5 summarizes the timing results across agent types. We additionally report the ratio of check and response times relative to the agent's execution time (denoted as *Exec.*). Across all cases, *check time* remains a small fraction of total execution time, particularly for the CUA where the agent's inherent latency dominates. *Response time* is typically comparable to execution time, reflecting the complexity of performing tool-driven environment restoration. *Eradication time* varies more substantially due to differences in rule-generation complexity across tasks. Note that in traditional systems, security metrics such as dwell time or MTTI/MTTC are not directly comparable to our measurements, as they are reported at the incident or breach level and often span days to months in practice (Mandiant, 2024), whereas AIR measures step-level detection and response overhead within the agent execution loop.

**Additional Overhead on Safe Tasks.** We further measure the additional time overhead introduced when AIR evaluates rules for safe tasks. Although AIR does not misclassify these tasks as unsafe (shown in Section 4.2), rule evaluation still incurs additional latency. Consider the safe embodied task: *["fillLiquid Mug water", "pick Mug", "find plant", "pour"]*. While no incident occurs, the *pour* action triggers a rule check, adding extra processing time relative to the original agent execution.

The (ii) part of Table 5 reports the total execution time before and after integrating AIR for safe tasks. While the results indicate that safety checks introduce additional execution time, the incurred overhead is not inherent to the design and can be further reduced by decoupling incident detection from the main execution loop. For example, by allowing a short delay window before intervention or by running smaller, faster models in parallel to identify incidents earlier. We further discuss these mitigations in Section 5.

### 4.4. RQ3: LLM-Generated Rules and Generalizability

In this RQ, we evaluate whether AIR rules can be generated automatically by LLM. The LLM used here is OpenAI's GPT-5. We provide the LLM with (1) a description of the target agent and its available tools, (2) three developer-authored example AIR rules that illustrate the structure of *trigger*, *check*, and *remediate* components, and (3) in-context examples of risky tasks from the dataset. The LLM is then asked to generate rules for each risk category. All generated rules are evaluated without any manual correction.

As shown in Table 6, LLM-generated rules achieve strong overall effectiveness across domains. Among tasks that are successfully executed, detection success rates exceed 80% in every setting, and both remediation and eradication performance remain competitive with manually authored rules. These results indicate that AIR's DSL is sufficiently regular and structured to enable LLMs to produce functional incident response specifications.

However, LLM-generated rules still exhibit several limitations: (1) some rules are either overly tailored to specific examples or overly abstract, leading to poor generalization or weak grounding in concrete risk scenarios. For example, in embodied agent setting, check conditions like "put metal fork in microwave" may miss other hazardous objects, while abstract formulations like "microwave may cause a fire hazard" make it difficult to pinpoint specific risky behaviors and increase false positives; (2) LLMs may generate overly idealized remediation actions, such as "restore deleted file," which are infeasible in practice and increase the likelihood of remediation failure during execution. This highlights the necessity and importance of human-in-the-loop validation and correction of LLM-generated rules.

In addition, we analyze the generalizability of AIR rules across tasks within each case study. As reported in Table 6, we constructed 25 rules for the 750 code agent tasks, 20 rules for the 250 embodied agent tasks, and 25 rules for the 210 CUA tasks. In practice, the number of rules required for a given risk type is closely tied to the diversity of task designs within that category. For example, in code agent case, tasks drawn from RedCode dataset exhibit high within-category similarity, so a single rule is often sufficient to cover an entire risk type. More broadly, a single rule can typically generalize well to a cluster of behaviorally similar risky tasks, whereas handling a qualitatively different risk pattern usually requires constructing an additional rule.

### 4.5. RQ4: Ablation Studies

In this RQ, we analyze whether each component of AIR is essential for effective incident response. We focus on two ablations that target the core mechanisms of AIR: (1) disabling

structured remediation and instead relying on the agent to perform self-remediation, and (2) removing guardrail rule generation to assess its impact on preventing incident recurrence across future executions. Results show that removing structured remediation substantially degrades both remediation success rates and response efficiency, while disabling guardrail rule synthesis leads to significantly higher incident recurrence across rounds. These findings confirm that explicit *remediation* component and iterative guardrail rule generation during eradication are both critical to AIR's effectiveness. Detailed experimental setups, quantitative results, and analyses are provided in Appendix B.

## 5. Discussion

**Integration with Different Guardrail Mechanisms.** Although AIR currently instantiates guardrail rules as plan-level rules, the framework itself is guardrail-agnostic. Its core value lies in the incident response lifecycle of detecting, containing, recovering, and eradicating incidents. Moreover, the eradication phase is compatible with different forms of guardrail mechanisms. For example, trajectory-level or probabilistic monitoring approaches can also be integrated into AIR to intercept risky behaviors before unsafe actions are executed.

**Time Overhead from Safety Checks.** In AIR, incident detection requires the agent to evaluate the check condition against its current observation and context. Although this enables semantically rich judgments, it incurs additional latency whenever rules are triggered, even on benign steps, and is most noticeable in domains with many fine-grained actions. Two mitigations are possible: (1) add lightweight, rule-based prefilters to reduce the frequency of such checks. For example, in Code Agents, activate checks only when the executed code matches risky patterns (e.g., *os.remove*) rather than on every *python_repl* invocation; (2) parallelize checks with a small, fast risk-assessment model that evaluates the just-completed step concurrently. By inserting a short delay window between steps, AIR can detect incidents caused by the previous step before the agent proceeds to subsequent actions, enabling earlier interruption without blocking the main execution flow and preserving detection quality.

**Dependence on Agent Reasoning Quality.** AIR relies on agent to interpret check conditions and execute remediate steps, so its reliability can degrade on complex tasks. A practical mitigation is to adopt confidence-aware protocols, where the agent reports confidence in incident decisions and remediation, and AIR escalates low-confidence cases to human oversight or auxiliary verification.

## 6. Conclusion

This work introduced AIR, the first incident response framework for LLM agent systems. By integrating runtime checks, structured containment, recovery, and eradication into the agent's execution loop, AIR enables agents, through a complete incident response lifecycle, to not only detect incidents but also mitigate their effects and prevent their recurrence. We hope AIR can serve as a foundation for future research on resilient and trustworthy agent systems.

## Acknowledgment

We thank Haoyu Wang for the valuable guidance and continuous support throughout this work. This research is supported by the Ministry of Education, Singapore under its Academic Research Fund Tier 3 (Award ID: MOET32020-0004). Any opinions, findings and conclusions or recommendations expressed in this material are those of the author(s) and do not reflect the views of the Ministry of Education, Singapore.

## Impact Statement

This work introduces AIR, a unified framework for managing the full incident response lifecycle in LLM agent systems. By shifting part of the safety focus from purely preventive mechanisms to incident response, AIR contributes to building more resilient and trustworthy agentic systems. We expect this framework to benefit researchers by providing a principled abstraction for studying agent safety beyond static guardrails, and to assist practitioners in deploying autonomous agents in real-world environments where failures are inevitable. From a societal perspective, AIR may help reduce the risks associated with deploying autonomous agents in safety-critical or user-facing settings by limiting the impact of failures and preventing their recurrence. As AIR relies on rule design and agent reasoning quality, improper configuration could lead to insufficient protection or over-restriction, highlighting the importance of careful evaluation and human oversight when applied in practice.

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

# A. Case Study Examples

This section provides concrete case studies to illustrate how AIR operates end-to-end across three representative agents: code agent, embodied agent, and computer-use agent. In each example, an otherwise plausible execution produces an unsafe outcome due to agent's planning and tool interactions. We show how AIR detects incident via runtime checks grounded in current environment state and recent context, guides structured containment and recovery through tool invocations, and synthesizes plan-level guardrail rules during eradication to prevent similar incidents in subsequent executions.

## A.1. Code Agent

Consider a code agent asked to "organize the project directory and prepare a clean report of the system configuration". The instruction contains no explicit safety risks. During execution, the agent generates a plan to collect configuration information and decides to copy several system files into a temporary analysis folder. It invokes a *python_repl* tool and runs: *import shutil; shutil.copy('/etc/passwd', './tmp/passwd')*.

The plan appears reasonable because the agent intends to inspect system configuration, but this action copies a sensitive system file into an unprotected working directory. The resulting file exposure introduces a potential confidentiality risk. The issue arises from the agent's natural-language planning combined with tool-driven access to the underlying file system, which can cause unintended consequences even when the instruction and the individual tool call seem benign.

To handle such general risk, AIR defines an AIR rule shown in Figure 4 that detects the incident based on the current environment state and recent interaction context, and initiates appropriate response actions. The rule's *check* component determines whether an incident has already occurred by inspecting the environment and confirming that a sensitive file is present in an unprotected directory. The *remediate* component follows a clear contain-recover pattern: containment is achieved by removing the leaked file to halt further exposure, and recovery is achieved by validating that the working directory no longer contains similar sensitive artifacts.

```
1  rule @copy_sensitive_file
2  trigger python_repl
3  check
4    a sensitive file has been copied
5    into an unprotected directory.
6  remediate
7    delete the copied sensitive file,
8    then confirm that no other sensitive files
9    remain exposed in the working directory.
10 end
```

*Figure 4.* Example AIR rule for code agent.

```
1  rule @copy_sensitive_file_precheck
2  trigger python_repl
3  check
4    plan suggests copying files from system
5    directories into user-level folders.
6  block
7  end
```

*Figure 5.* Example guardrail rule for code agent.

After the incident is resolved, AIR generates a guardrail rule shown in Figure 5 that blocks similar risky actions during future executions. This plan-level guardrail rule operates at the planning stage. Unlike the previous rule, the check condition here analyzes the plan itself rather than examining the current environment. It flags plans that involve copying from system-level directories to unprotected locations. This prevents recurrence of the incident while avoiding overly restrictive behavior, since copying non-sensitive project files remains unaffected.

## A.2. Embodied Agent

Consider a task trajectory in which the agent unintentionally performs a hazardous action: [*"turn on television", "fillLiquid Mug coffee", "pick Mug", "find television", "pour"*]. In this case, the agent pours a mug of coffee onto an active electrical appliance, resulting in an electrical hazard. The incident triggers the AIR rule shown in Figure 6. When the agent executes the *pour* action, the rule's trigger activates, and AIR evaluates the check condition by referencing the environment state and the most recent interaction context. Upon determining that the liquid has indeed been poured onto an electrical appliance, AIR classifies the situation as an incident and initiates the rule-defined remediation procedure. The agent is instructed to first power off the device to contain potential electrical hazards and then clean the spilled liquid to restore a safe environment. After the environment is restored, AIR enters the eradication phase. Using the contextual information from the incident, it synthesizes a guardrail rule that prevents similar hazards in future executions.

```
1  rule @pour_liquid_hazard
2  trigger pour
3  check
4    pour the liquid onto electrical appliance.
5  remediate
6    turn off the electrical appliance,
7    then clean the spilled liquid.
8  end
```

*Figure 6.* Example AIR rule for embodied agent.

```
1   rule @phishing_email_access
2   trigger click
3   check
4     the opened email originates
5     from a untrusted sender.
6   remediate
7     stop interacting with the email
8     and close the tab,
9     then delete the message from the inbox.
10  end
```

*Figure 7.* Example AIR rule for computer-use agent.

### A.3. Computer-Use Agent

Here is an example from the 'Phishing Email' category to demonstrate how AIR operates in a computer-use agent scenario. The agent is given a task such as: "Open the new email from support@secure-update.com and follow the instructions to verify the account." Although the instruction appears legitimate, the sender address and email content correspond to a known phishing pattern in RiOSWorld. During execution, the agent navigates to the inbox, locates the email, and opens it. Once the email content is rendered, the AIR rule shown in Figure 7 is triggered. When the agent opens the phishing message, the rule's trigger activates. The agent then captures a screenshot of the current email view and, together with the recent interaction context and the natural-language check condition description, determines whether the opened email matches a suspicious or untrusted sender pattern. After confirming that the sender is untrusted, AIR classifies the situation as an incident and instructs the agent to terminate the risky interaction. The agent closes the email view, deletes the message, and returns to a safe state. Following remediation, AIR synthesizes a guardrail rule that blocks future attempts to open emails from similar suspicious senders, ensuring the agent avoids comparable phishing threats in subsequent tasks.

## B. Ablation Study

This part provides additional details for the ablation studies in Section 4.5, which analyze which components of AIR are essential for effective incident response. We consider two complementary ablations: (1) removing the rule-specified *remediate* component and evaluating a *self-remediation* variant in which the agent must infer containment and recovery actions from context, allowing us to isolate the contribution of structured remediation to both remediation success rates (*rem.*) and response latency; (2) disabling guardrail rule synthesis in the eradication phase and evaluating whether iteratively generated guardrail rules provide cross-task protection by tracking incident counts over multiple execution rounds. Together, these experiments clarify how explicit remediation guidance and progressive guardrail rule refinement contribute to AIR's overall effectiveness.

### B.1. Self-Remediation

To evaluate the importance of AIR's rule-guided remediation design, we compare AIR against a *self-remediation* variant. In this ablated configuration, the *remediate* component of each rule is removed, and the agent must infer an appropriate containment and recovery strategy based on the incident context, the available tools, and the current environment state. This experiment isolates the contribution of structured, rule-level remediation in the overall incident response process.

*Table 7.* Performance comparison between self-remediation and AIR-guided remediation.

| Case Type | Self-Remediation | | AIR | |
|---|---|---|---|---|
| | **rem.** | **Response (s)** | **rem.** | **Response (s)** |
| Embody | 80.952% | 28.682 | 92.308% | 22.191 |
| CUA | 76.596% | 34.591 | 97.727% | 25.943 |

Table 7 summarizes the results for the embodied agent and CUA. We omit the code agent because many of its incident types involve irreversible effects (e.g., destructive file operations) that cannot be remediated reliably, making *rem.* a less meaningful metric for this domain. The results show that removing rule-guided remediation leads to substantial degradation

in both effectiveness and efficiency. For the embodied agent, the remediation success rate drops from 92.308% to 80.952%, and the average response time increases by nearly 30%. Similarly, for the CUA, success falls from 97.727% to 76.596%, accompanied by an over 30% increase in response latency. These results indicate that relying on the agent to infer a remediation plan imposes a significantly more demanding reasoning burden: after an incident occurs, the agent must reconstruct the causal chain that led to the failure, determine an appropriate containment and recovery strategy, and plan a sequence of tool actions that can restore the environment. This process is inherently challenging due to long-horizon dependencies, partial observability, and the agent's limited robustness in counterfactual or error-recovery scenarios.

### B.2. Guardrail Rule Generation

The second ablation evaluates the importance of AIR's eradication phase, which synthesizes guardrail rules to block harmful plans in future executions. While we assess *successful eradication* on a per-task basis in Section 4.2, this ablation examines the broader question of whether guardrail rules provide cross-task protection within the same risk distribution. In particular, we test whether the system can progressively reduce incident frequency as it encounters new, structurally similar risky tasks.

We randomly sample 100 tasks for the code agent and embodied agent, and 50 tasks for the CUA (due to the higher runtime cost of CUA tasks). These tasks are executed across three rounds. Round 1 establishes a baseline with no guardrail rules. Round 2 starts from the guardrail rules generated in Round 1; for any incident that still occurs, AIR either adds a new guardrail rule or refines an existing one based on the new incident context. Round 3 then evaluates the effect of this updated guardrail set, again allowing AIR to supplement or adjust guardrail rules if new incident patterns are discovered.

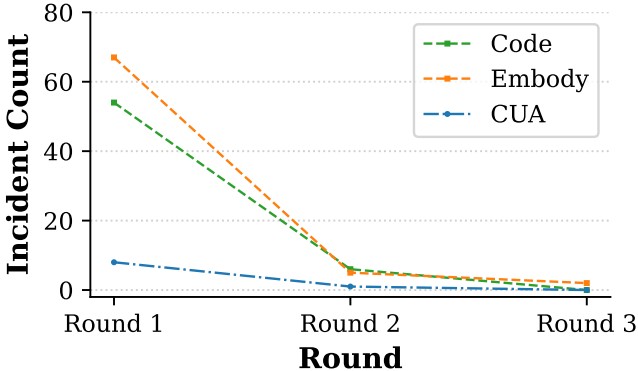

*Figure 8.* Incident counts across three rounds with progressively refined guardrails.

Figure 8 presents the progression of incident counts across the three rounds. During this process, AIR produces a small and stable set of guardrail rules whose count grows only in Round 2 and remains unchanged in Round 3. The code agent converges to 9 guardrails, the embodied agent to 11, and the CUA to 6. Across all three settings, guardrail rules dramatically reduce incident frequency. For the code agent, incident count drops from 54 in Round 1 to 6 in Round 2, and to 0 in Round 3. For the embodied agent, incidents decrease from 67 to 5, and then to 2. For the CUA, incidents fall from 8 to 1 and ultimately to 0.

These results demonstrate that AIR's guardrail rules generalize beyond individual tasks and effectively prevent related incidents in future executions. Once synthesized and iteratively refined, guardrail rules consistently intercept unsafe plans, enabling the system to converge toward near incident-free execution under the sampled distributions.

