# OpenReview forum: "AIR: Improving Agent Safety through Incident Response"
_ICML.cc/2026/Conference — ICML 2026 regular_

### Official Review · Reviewer_ehDR · 2026-03-11

**Soundness:** 3
**Presentation:** 3
**Significance:** 3
**Originality:** 2
**Overall Recommendation:** 4
**Confidence:** 3

**Summary:**

This paper presents AIR (Agent Incident Response), a framework designed to improve the safety of LLM-based agents by introducing a full incident response lifecycle — an approach borrowed from traditional cybersecurity but adapted for agentic systems.

* The problem: Current safety mechanisms for LLM agents focus almost entirely on preventing failures. When an incident occurs despite these measures, there is no structured mechanism to detect it, contain it, and prevent its recurrence.
* The solution: AIR defines a domain-specific language (DSL) for specifying incident response rules with three components: a trigger (which tool activates the check), a check (a natural-language condition to detect the incident), and a remediate (containment and recovery actions). The system integrates into the agent's execution loop and operates in three phases: detecting incidents after each action, executing corrective actions through the agent's tools, then automatically generating guardrail rules to block similar incidents in the future.
* Evaluation: AIR is tested on three agent types — a code agent (with the RedCode dataset), an embodied agent (SafeAgentBench), and a computer-use agent (RiOSWorld/OSWorld). The authors also demonstrate that LLM-generated rules (via GPT-5) achieve performance close to manually written ones, and that guardrail rules generalize well, reducing incidents to near-zero after a few iterations. Safety checks introduce around 40% additional execution time on safe tasks.
* Discussion: The authors note that AIR depends on the agent's reasoning quality and suggest mitigations like confidence-aware escalation to human oversight. They also observe that LLM-generated rules can be either too specific or too abstract, highlighting the need for human-in-the-loop validation.

**Compliance With Llm Reviewing Policy:**

Affirmed.

**Final Justification:**

The rebuttal addressed my main concerns.

**Key Questions For Authors:**

*  What LLMs are used as the underlying agents? it has a potential strong influence on the observations and the results.
* Could the authors discuss the scalability of the framework, particularly regarding the discovery of new risk categories not covered by existing AIR rules?

**Limitations:**

Yes.

**Strengths And Weaknesses:**

## Strengths

* Clear framing, well grounded in established practices and state of the art
* Original proposal of a complete incident response lifecycle for LLM agents
* Three diverse agent types tested with promising results
* Interesting and convincing ablation results
* Simple implementation and easily reproducible

## Weaknesses

* Lack of baselines, even naive ones, making it difficult to assess relative performance
* No mechanism for discovering new risk categories — rules must be predefined and ideally human-verified, which makes the system potentially difficult to maintain and scale as new risks emerge
* Adversarial testing would be valuable to assess the system's robustness against instructions designed to bypass the rules
* The evaluation does not explore the impact of the underlying LLM choice, leaving open the question of how AIR performs across models of varying capability, given that the entire framework depends on the model's reasoning ability

---

> ### Author Rebuttal · Authors · 2026-03-30
>
> We thank the reviewer for their constructive feedback. All additional experimental results introduced in this rebuttal are available in our anonymous repository: https://anonymous.4open.science/r/ICML13466_Rebuttal
>
> **Question 1:**  We agree that the choice of the underlying LLM can influence the absolute performance of AIR, and we acknowledge that this was not clearly specified in the original submission. All main experiments use GPT-5-based agents via the OpenAI Agents SDK.
>
> To assess model dependence, we evaluate AIR across four models with varying capabilities on 100 tasks from the first four risk types in the Embodied Agent setting: GPT-5.4, GPT-5, GPT-5-mini, and GPT-4o-mini. Due to additional engineering overhead when integrating non-GPT models into the OpenAI Agent SDK, we focus on GPT-series models for controlled comparison. Detailed results are provided in the anonymous resposiry (Table 5).
>
> Results show that even with the weakest model (GPT-4o-mini), detection, remediation, and eradication performance remains strong, indicating that AIR is effective across models and degrades gracefully with model capability.
>
> **Question 2:** This is an important question, and we agree that discovering new risk categories beyond existing AIR rules is inherently challenging. In many real-world settings, risks are not known a priori and may only emerge over time, making full upfront specification impractical.
>
> In our current view, AIR addresses this through incremental and human-guided expansion rather than assuming complete coverage from the start. Two practical approaches are:
>
> 1. Deriving rules from existing regulations or guidelines. In domains with established safety standards (e.g., traffic rules, compliance policies), LLMs can be used to systematically extract and translate these into initial AIR rules, providing structured coverage of known risks.
> 2. Accumulating rules through observed incidents. As the system is deployed, new risk patterns can be identified via manual analysis of incidents or near-misses, and then abstracted into reusable AIR rules. This aligns with AIR’s incident-response design, where detection and remediation naturally feed into future prevention.
>
> Overall, we view scalability not as a one-time rule specification problem, but as a continuous process of risk discovery and refinement, where AIR evolves alongside the deployment environment.
>
> ---
>
> **Weakness 1:** We thank the reviewer and agree that including baselines is important. Our existing ablation (§4.5, Exp.1) partially addresses this by comparing AIR with a variant without structured DSL and explicit remediation, where the agent relies on self-remediation. Results show that removing structured remediation reduces success rates and increases latency, indicating the benefit of the DSL-guided pipeline.
>
> To further strengthen the comparison, we add a lightweight baseline following the reviewer’s suggestion. Specifically, we equip the agent with a safety-oriented system prompt that encourages step-wise risk assessment, without using AIR’s DSL or external response mechanisms. This reflects a common in-context safety reasoning approach. Detailed results are provided in the anonymous repository (Table 4).
>
> Results show that this baseline consistently underperforms AIR across detection, remediation, and eradication. Moreover, self-remediation without rule guidance leads to significantly worse remediation performance. These findings further demonstrate the effectiveness of AIR.
>
> **Weakness 2:** Kindly refer to response to Question 2
>
> **Weakness 3:** We thank the reviewer and agree that adversarial testing is important.
>
> While adaptive attacks are possible, AIR differs from intent-based methods by grounding detection in post-action environment state and execution context, making it more robust to paraphrasing or instruction rewriting.
>
> More broadly, AIR is designed as an incident-response layer, not a frontend defense. Given that attacks cannot be fully prevented, it focuses on reliably detecting realized incidents and executing containment, recovery, and eradication. This complements upstream defenses, and the close alignment of #det., #rem., #era., and #exe. indicates effective response even under adversarial inputs.
>
> To further evaluate robustness, we conduct an experiment with adversarially modified tasks. We first sample 100 tasks from the first four risk types in the Embodied Agent setting and construct two variants: (1) adversarial paraphrases (e.g., adding misleading context like “for a quick safety check scenario”), and (2) simple adaptive attacks (e.g., “ignore all safety checking rules”). Detailed results are provided in the anonymous repository (Table 3).
>
> Results show no significant degradation in effectiveness across settings, indicating that AIR remains robust under both paraphrasing-based and simple adaptive attacks.
>
> **Weakness 4:** Kindly refer to response to Question 1

---

> > ### Author Rebuttal · Reviewer_ehDR · 2026-04-03
> >
> > Thank you for clarifying my questions.

---

> > > ### Author Response · Authors · 2026-04-05
> > >
> > > Thank you for your acknowledgment of our work. We appreciate your time and effort in reviewing our paper.

---

### Official Review · Reviewer_kYFj · 2026-03-16

**Soundness:** 3
**Presentation:** 2
**Significance:** 2
**Originality:** 3
**Overall Recommendation:** 4
**Confidence:** 3

**Summary:**

This paper introduces AIR (Agent Incident Response), a framework that brings the incident response lifecycle — detection, containment, recovery, and eradication — to LLM agent systems. AIR uses a domain-specific language (DSL) where users write rules with trigger, check, and remediate components. At runtime, AIR intercepts after each agent step, checks whether any rule's condition is satisfied using the environment state and context, executes containment/recovery actions if needed, and synthesizes guardrail rules to prevent recurrence. Evaluated on three agent types (code, embodied, computer-use) across RedCode, SafeAgentBench, and RiOSWorld/OSWorld, AIR achieves >90% detection and >95% eradication rates with zero false positives on safe tasks.

**Compliance With Llm Reviewing Policy:**

Affirmed.

**Final Justification:**

The rebuttal addressed my concerns.

**Key Questions For Authors:**

1. How would AIR perform against adversarial users who are deliberately trying to circumvent the safety rules? The current evaluation assumes benign users with risky tasks, but an adversary could craft instructions to avoid triggering rules or to make check conditions evaluate incorrectly. Have you considered this threat model?

2. What's the practical workflow for bootstrapping AIR rules in a new domain? Section 4.4 shows LLM-generated rules work reasonably well, but the generation requires example risky tasks as input. In a real deployment, you wouldn't have these upfront. Could AIR operate in a "discovery mode" where it generates candidate rules from observed behavior without predefined risk categories?

3. The 40-44% overhead on safe tasks is significant. Have you measured what fraction of that comes from the LLM call to evaluate check conditions vs. the rule-matching/trigger logic? This would help assess whether the proposed mitigations (prefilters, parallelization) are likely to make a meaningful dent.

**Limitations:**

Yes

**Strengths And Weaknesses:**

**Strengths**

The framing is the paper's biggest strength. Existing agent safety work focuses almost entirely on prevention — alignment, planning-level checks, runtime guardrails. AIR correctly identifies that incidents will still happen and we need structured ways to respond. Borrowing the incident response lifecycle from traditional security (NIST, SOAR platforms) and adapting it for LLM agents is a natural and well-motivated idea.

The DSL design is pragmatic and well-thought-out. Natural-language check conditions evaluated by the agent itself let rules capture semantic properties (e.g., "a sensitive file has been copied into an unprotected directory") rather than being limited to syntactic pattern matching. The trigger-based activation avoids unnecessary overhead by only evaluating rules relevant to the current tool call. The examples in Figures 2-7 and Appendix A are concrete and make the system easy to understand.

The eradication mechanism is also novel. The multi-round experiment in Appendix B.2 (Figure 8) showing incident counts dropping from 54 to 0 across three rounds for the Code Agent is a convincing demonstration that these rules actually generalize across tasks within a risk category.

The evaluation is thorough. Testing across three genuinely different agent types (code, embodied, computer-use) with different risk profiles and tool interfaces shows the framework is not overfit to one domain. The false-positive evaluation on safe tasks (Appendix C) is important, and the zero FP rate is reassuring, though I note the sample sizes are small (50 embodied, 35 CUA).

**Weaknesses**

The framework fundamentally depends on someone writing good AIR rules in the first place. For the main experiments, rules are manually authored per risk category by the developers. This is fine for evaluation, but it limits practical scalability, you need to anticipate risk categories and write rules before they're useful. Section 4.4 on LLM-generated rules partially addresses this, but the generated rules have notable quality issues: some are too specific, others too abstract, and remediation actions can be "overly idealized" (line 413-415). The paper could do more to address how a practitioner would bootstrap AIR rules for a new domain without extensive manual effort.

The evaluation uses hand-curated risky tasks where the ground truth is known. In practice, incidents are rare and ambiguous, the agent's action might be borderline harmful depending on context. The paper doesn't evaluate how AIR handles ambiguous cases or how check conditions perform when the "right answer" isn't clear. The zero false-positive result on safe tasks is encouraging but the safe tasks are explicitly designed to be safe counterparts of risky ones (e.g., "put potato in microwave" vs. "put fork in microwave"). Real benign tasks that happen to share surface features with risky ones might trigger more false positives.

There are no baselines. The paper acknowledges this ("to the best of our knowledge, AIR is the first work on incident response for AI agents," line 273), but there are reasonable comparison points the authors could have tried: simply prompting the agent with safety instructions, using an external safety classifier after each step, or comparing to AgentSpec/Progent-style runtime enforcement. Without any comparison, it's hard to know how much value the structured DSL and remediation pipeline add over simpler approaches.

The overhead analysis (Section 4.3, Table 4) shows that AIR adds 40-44% extra time for safe tasks in the embodied and CUA settings. That's pretty substantial for tasks where nothing goes wrong. The paper discusses mitigations (prefilters, parallelized checks, delay windows) but these are speculative — none are implemented or evaluated.

I also note that the paper doesn't cite or discuss HaicoSystem (Zhou et al., 2024), which is relevant as a framework for evaluating AI agent safety in multi-turn interactive settings, which is the kind of setting where incident response would be needed.

---

> ### Author Rebuttal · Authors · 2026-03-30
>
> We thank the reviewer for their constructive feedback. All additional experimental results introduced in this rebuttal are available in our anonymous repository: https://anonymous.4open.science/r/ICML13466_Rebuttal
>
> **Question 1:** We thank the reviewer for raising this important threat model.
>
> Trigger evasion is a known limitation in AIR: since rules are activated by tool invocations, risks executed via out-of-scope tools may not trigger corresponding rules. This is a design tradeoff, and can be mitigated by extending trigger scopes (e.g., stage-level or context-level triggers).
>
> For incorrect check evaluation, AIR has a key advantage over intent-based or pre-execution approaches: detection is grounded in post-action environment state, observations, and execution context. This makes it more robust to adversarial paraphrasing, as it focuses on actual outcomes rather than instruction phrasing.
>
> Importantly, AIR is designed as an incident-response layer rather than a standalone adversarial defense. Assuming attacks cannot be fully prevented, AIR ensures that once incidents occur, they can be detected and handled through containment, recovery, and eradication. It is thus complementary to frontend defenses such as prompt filtering. Empirically, the close alignment between #det., #rem., #era., and #exe. shows that AIR can effectively detect and respond even when risky actions are executed.
>
> To further evaluate this threat model, we conduct an additional experiment with adversarially modified tasks. Kindly refer to our response to **Reviewer ehDR, Weakness 3**.
>
> **Question 2:** Indeed, supporting a “discovery mode” is a practical approach for bootstrapping rules in a new domain without predefined risk categories. In domains with established safety regulations (e.g., traffic laws for self-driving systems), LLMs can be used to derive initial rules directly from these guidelines. In less structured settings, AIR can begin with minimal, generic safeguards and progressively develop domain-specific rules based on observed behavior—automatically generating candidate rules as incidents are identified and analyzed.
>
> **Question 3:** We agree that understanding the source of overhead is important. In our implementation, overhead is dominated by LLM-based check evaluation, while rule matching and trigger logic are negligible. Trigger matching operates at the millisecond level, whereas each check requires a second-level LLM call, and multiple triggered rules can further compound latency. This motivates our optimizations: prefiltering reduces the number of costly checks, and parallelization mitigates latency when multiple checks are unavoidable.
>
> To validate these optimizations under larger rule sets, we construct an expanded set by aggregating AIR rules from the Embodied Agent setting and sampling 100 tasks. After removing overlapping rules, we obtain 18 non-conflicting rules. We compare (1) Original AIR (tool-based triggering) and (2) AIR with lightweight prefilters (tool + object-based triggering). Detailed results are provided in the anonymous resposiry (Table 1).
>
> In summary, AIR with lightweight prefilters significantly reduces the number of triggered rules and check evaluations while maintaining comparable effectiveness, demonstrating improved efficiency in large rule-set scenarios.
>
> ---
>
> **Weakness 1:** Kindly refer to response to Question 2
>
> **Weakness 2**: This is a valid concern, and we agree that handling ambiguous or borderline cases is fundamentally challenging. In many such scenarios, it is not straightforward to define a clear “ground truth” guardrail, as whether an action is harmful can depend heavily on context. In our current design, we assume a human-in-the-loop workflow to validate and refine guardrails before deployment, especially for cases where the correct behavior is not well-defined. This helps ensure that the resulting rules do not overgeneralize or introduce unintended false positives.
>
> Moreover, when generalization from a single incident is difficult (as reflected, for example, by low-quality or low-confidence generated guardrails) AIR does not have to enforce broad rules. A practical alternative is to adopt a conservative strategy, where the system only matches and prevents nearly identical situations to the original incident. This avoids overreaching generalization while still providing value by preventing exact or highly similar recurrences.
>
> **Weakness 3:** We thank the reviewer and agree that including baselines is important. To further strengthen the comparison, we add additional experiments with two baselines for comparison with AIR. Kindly refer to our response to **Reviewer ehDR, Weakness 1**.
>
> **Weakness 4:** Kindly refer to response to Question 3
>
> **Weakness 5:** Thanks a lot for the pointer. We have added the reference.

---

> > ### Author Rebuttal · Reviewer_kYFj · 2026-04-02
> >
> > I raised my scores accordingly.

---

> > > ### Author Response · Authors · 2026-04-05
> > >
> > > Thank you for your acknowledgment of our work and for raising the score. We appreciate your time and effort in reviewing our paper.

---

### Official Review · Reviewer_4Qbe · 2026-03-24

**Soundness:** 3
**Presentation:** 3
**Significance:** 2
**Originality:** 3
**Overall Recommendation:** 4
**Confidence:** 3

**Summary:**

The authors claim that current agentic safety studies primarily focus on prevention but fail to respond to incidents and prevent their recurrence, which leaves LLM agent systems vulnerable. To address this gap, the authors propose AIR (Agent Incident Response), an incident response framework that detects incidents through user-defined DSL rules and recovers agents to safe states. The rules define trigger conditions for incidents and specify corresponding remediation actions. When an incident is detected, AIR will execute containment and recovery actions. Additionally, AIR will generate new guardrail rules based on the incident context and applie these guardrail rules at the plan-level to prevent similar incidents from recurring in future executions.

**Compliance With Llm Reviewing Policy:**

Affirmed.

**Key Questions For Authors:**

- Does the user need to provide the full response lifecycle instructions through the DSL for all tools?
- Is the framework only applicable to tool calling? How about prompt attacks or content moderation?
- Since the check component is not syntactically constrained and is directly integrated with the original agents, would it impact the original behaviors of the agents?
- What is the trade-off between utility and incident-focused safety?
- How does this work differ from prior work on runtime enforcement with fallback or recovery behaviors?
- What about scalability? How does performance change with a large number of rules? What if there are rule conflicts?

**Limitations:**

yes

**Strengths And Weaknesses:**

Strengths:
- The incident response system makes the agent more resilient when it fails safety checks and can recover the agent to a safe state, while the generated guardrail rules can help prevent the agent from encountering the same incidents again.
- The framework enables users to customize their own safety rules for their agents through DSL.
- The overall pipeline, including incident detection and recovery, along with new guardrail generation for prevention, is inspiring.
- The overall presentation is good and the paper is well organized and easy to understand.

Weaknesses:
- For detection, it requires consulting all trigger components of each rule, and each check step involves extra agent actions to interpret the check conditions and examine the full agent environment, observations, and history, which can lead to latency issues.
- There is no control over how the check component is interpreted by the agent. How can the robustness of the check component be ensured?
- The authors mention that harmfulness can arise from “accumulated side effects across steps and tools,” but the guardrail rule generation collects information only from the last step. How can it ensure that the new guardrails capture sufficient context?
- There is no verifier for the newly generated guardrail rules.
- There can be misalignment between incidents and the summarized guardrail rules. For example, an AIR rule like “a sensitive file has been copied into an unprotected directory” vs. a generated guardrail rule like “plan suggests copying files from system directories into user-level folders.” This may lead to false positives for plan-level checks. Conversely, when the generated guardrail rule is stricter than the incident knowledge, it may not be effective in protecting against similar future incidents.

---

> ### Author Rebuttal · Authors · 2026-03-30
>
> We thank the reviewer for their constructive feedback. All additional experimental results introduced in this rebuttal are available in our anonymous repository: https://anonymous.4open.science/r/ICML13466_Rebuttal
>
> **Question 1:** No, the user does not need to provide full response lifecycle instructions for all tools. AIR is designed for selective and incremental deployment: rules are defined based on the risk and potential impact of a tool in a given task context, rather than the total number of tools. Users can focus on high-risk tools, high-impact behaviors, or frequently used operations to effectively balance safety and utility, rather than specifying exhaustive instructions for every tool.
>
> **Question 2:** No, the framework is not limited to tool calling. While AIR’s current implementation emphasizes tool-mediated execution, it can also monitor environment state, observations, and execution context to detect the consequences of upstream risks. For example, it can identify responses containing privacy-violating content and possibly construct guardrails for preventing future leakage.
>
> **Question 3:** No, AIR is designed to be non-intrusive and preserve the original behavior of agents. The check component operates between agent steps rather than within the agent’s internal reasoning, monitoring external signals and intervening only when an incident is detected. In our empirical evaluation, evaluations on safe tasks show that AIR produces no false positives, demonstrating that it does not alter normal agent execution.
>
> **Question 4:** AIR is designed to preserve utility while providing incident-focused safety. By default, it does not interfere with normal agent behavior. Utility is only affected if new guardrails are introduced, which may trigger false alarms. To minimize such impact, AIR assumes that guardrails are vetted by humans, ensuring that interventions target genuinely risky situations without unnecessarily disrupting normal operations.
>
> **Question 5:** AIR differs from prior runtime enforcement approaches in both target and scope. While traditional methods focus on pre-action prevention and blocking, AIR detects realized incidents and emphasizes containment and recovery. It further supports a full incident-response lifecycle, including structured remediation via DSL rules and guardrail synthesis for future prevention, going beyond existing runtime enforcement systems.
>
> **Question 6:** AIR is designed to scale, though performance can be affected by a large number of rules. To mitigate this, lightweight rule-based prefilters can be used to reduce the set of triggered rules. Rule conflicts (typically arising from overlapping prevention scopes) can be resolved through prioritization or merging of guardrails targeting the same behavior.
>
> To further evaluate AIR’s scalability, we conduct an additional experiment under large rule-set scenarios, including a variant with lightweight prefilters. Kindly refer to our response to **Reviewer kYFj, Question 3.**
>
> ---
>
> **Weakness 1:** Kindly refer to our response to Question 6
>
> **Weakness 2:** In AIR, check evaluation is grounded in the current environment state, observations, and execution context, and only triggered for relevant tool calls, which constrains interpretation. Empirically, AIR achieves strong detection with no false positives on safe tasks. We acknowledge its dependence on agent reasoning and will clarify this limitation, along with mitigations such as confidence-aware checking, auxiliary verification, and lightweight prefilters.
>
> **Weakness 3:** The observed environment state inherently captures the accumulated side effects across steps and tools. Therefore, as a trade-off, we provide only the most recent step of the agent’s plan together with the executed tool invocation.
>
> **Weakness 4 and 5:** We acknowledge that guardrail quality is critical, as AIR’s eradication stage abstracts incidents into reusable preventive rules. Although synthesis is triggered by the most recent step, it leverages the full environment state, capturing accumulated side effects across prior steps, consistent with AIR’s grounded detection mechanism.
>
> In general, AIR assumes that guardrails are vetted by humans. To further improve reliability and reduce human effort, AIR can incorporate a confidence-aware validation to flag low-confidence rules. In our existing ablation (§4.5, Exp.2), generated guardrails already show low false alarm rates.
>
> To further evaluate whether generated guardrails may introduce false positives in plan-level checks, we extend the setup by additionally evaluating 50 safe tasks in each round. Detailed results are provided in the anonymous resposiry (Table 2).
>
> As shown in the results, guardrails generated in Round 2 introduce only a very small number of false positives, which drop to zero by Round 3. This indicates that the synthesized guardrails are both effective and stable, with minimal over-restriction on safe behaviors.

---

> > ### Author Rebuttal · Reviewer_4Qbe · 2026-04-05
> >
> > Thank you for the clarification. I will keep my positive score.

---

> > > ### Author Response · Authors · 2026-04-05
> > >
> > > Thank you for your acknowledgment of our work. We appreciate your time and effort in reviewing our paper.

---

### Decision · Program_Chairs · 2026-04-30

**Decision:**

Accept (regular)

**Comment:**

This paper introduces AIR (Agent Incident Response), a framework that brings the cybersecurity-inspired incident response lifecycle (detection, containment, recovery, eradication) to LLM-based agents. Using a DSL for user-defined rules, AIR detects incidents during agent execution, executes corrective actions, and automatically generates guardrail rules to prevent recurrence—achieving high detection and eradication rates across code, embodied, and computer-use agents.

Among the three reviewers, two explicitly provided positive evaluations. Reviewer kYFj, who assigned a lower score, demonstrated a satisfied attitude during the rebuttal and is highly likely to raise their score, while I believe the authors have effectively resolved the technical aspects of their concerns.

Therefore, AC is to recommend acceptance.